# A Graph-Based Representation Learning Approach for Breast Cancer Risk Prediction Using Genotype Data

## Abstract

Breast cancer risk prediction using genotype data is a critical task in personalized medicine. However, the high dimensionality and potential redundancy of genetic features pose challenges for accurate risk prediction. We present a graph-based representation learning pipeline for breast cancer risk prediction. Our method addresses the issue of feature redundancy by developing an ensemble-based feature selection approach. We evaluated the performance of the graph-based approach in a breast cancer risk prediction task using a dataset of 644,585 genetic variants from Biobank of Eastern Finland, consisting of 168 cases and 1558 controls and compared it with the classical machine learning models. Using 200 top-ranked genetic variants selected by the ensemble approach, the graph convolutional network (GCN) achieved area under the ROC curve (AUC) of $0.986 \pm 0.001$ in discriminating cases and controls, which is better than an XGBoost model with AUC of $0.955 \pm 0.0034$.

## 1 Introduction

Breast cancer is a significant global health concern with 2.3 million new diagnoses and 685,000 deaths reported in 2020. Estimating breast cancer risk involves considering a range of factors that contribute to an individual's likelihood of developing the disease. Traditional methods for estimating breast cancer risk often encompass known factors related to personal and family medical history, genetics, lifestyle choices, and hormonal influences. However, such models use a significant amount of time and have varied limitations Gail et al. (1989); Tice et al. (2008).

Recent advancements in breast cancer risk prediction models have incorporated genetic information, specifically single nucleotide polymorphisms (SNPs) to distinguish between individuals affected by breast cancer and those who are healthy Tao et al. (2023); Behravan et al. (2018); Gao et al. (2022); Ahearn et al. (2022); Ho et al. (2020). SNPs are the most common type of genetic variation, occurring when a single nucleotide at a specific location in the genome differs among individuals. Each SNP typically has two alleles, like A or T, defining the genetic variation at that position. In a person's genome, there are approximately 4 to 5 million SNPs. In genome-wide association studies (GWAS), researchers have found genetic variants strongly linked to breast cancer Jia et al. (2022); The BioBank Japan Project et al. (2022). The typical approach involves testing each single genetic variant's association with the disease by comparing the frequencies of alleles/genotypes between affected individuals and healthy controls. However, this method overlooks potential correlations or interactions among multiple genetic variants (SNPs) as it focuses on one SNP at a time. Considering all SNPs together becomes challenging due to the large number of genetic variants, complex interactions among them, and often limited sample size in the study.

There is a scarcity of existing research in the field of utilizing machine and deep learning models for predicting the disease risk through SNP data modeling. This scarcity arises from both the complex characteristics of the high-dimensional SNPs data and the challenges associated with acquiring sensitive genomics information. To tackle the challenge of high dimensionality, current methodologies often suggest employing feature selection techniques as a preliminary step. These methods aim to first reduce the dimensionality of SNP data before utilizing the resulting lower-dimensional features for subsequent tasks Pudjihartono et al. (2022). For instance, Alzoubi et al. (2023) employed

a multilayer perceptron (MLP) to sequentially identify hundreds of relevant SNPs. This was done to forecast the disease status of individuals within a case and control cohort. Tai & Dhaliwal (2022) conducted a comparison between an MLP and several conventional machine learning models. This comparison was based on the utilization of 104 malaria-related SNPs collected from the existing literature. Meanwhile, Behravan et al. (2018) introduced an iterative strategy based on a gradient boosting algorithm. They employed this approach to filter out SNPs linked to breast cancer risk, followed by the use of a support vector machine (SVM) classifier for breast cancer risk prediction. Despite achieving promising results, these models are challenging to interpret. The majority of these techniques utilize linear feature selection approaches. They select representative features by ranking their corresponding feature weight vectors. However, these operations often treat each feature in isolation and overlook the intricate higher-order SNP relationships that exist among the original features. Consequently, such approaches often lead to redundancy among the selected features.

In this study, we advance the breast cancer case-control prediction using graph-based deep learning architectures. We address the challenge of SNPs filtering by employing an ensemble-based feature selection approach to efficiently capture non-linear and high-dimensional SNP-SNP interactions. We have used three distinct graph neural networks: GCN Morris et al. (2019), graph attention network (GAT) Veličković et al. (2017), and graph sample and aggregation network (GraphSAGE) Hamilton et al. (2017) for the classification of breast cancer case-control using the top $K$ SNPs filtered using our proposed feature selection approach. To demonstrate the efficacy of the graph-based models, we conducted comparative experiments with widely-used machine learning classifiers such as XGBoost, random forest, SVM classifier, and logistic regression.

## 2 MATERIALS

We used genotype data from 168 breast cancer cases and 1558 controls from the Biobank of Eastern Finland. Detailed procedures for genotyping, allele calling, and quality control were followed as outlined in Kurki et al. (2022). All samples were collected with informed written consent based on the Finnish Biobank Act. For quality control, we filtered the SNPs using PLINK Purcell et al. (2007) software. SNPs with missing variants lower than 5%, minor allele frequency of $\leq 0.005$, and Hardy-Weinberg equilibrium (HWE) with log $p$-values $\leq 5$ were excluded. Finally, we kept the SNPs with linkage disequilibrium of $r^2 \leq 0.6$, leading to 644,585 total SNPs for the breast cancer risk prediction task, in this study. For the original encoding of SNPs, an additive scheme was employed Mittag et al. (2015). This entails representing each SNP based on the count of minor alleles, where homozygous major, heterozygous, and homozygous minor genotypes are encoded as 0, 1, and 2, respectively.

## 3 METHODS

### 3.1 ENSEMBLE-BASED SNPs SELECTION

We trained an ensemble-based neural network (ENN) to aggregate multiple feature selection methods. Figure 1 illustrates the proposed ENN model. For every individual SNP $i$, an independent feature selection method computes its importance score $S_i$. Consider four distinct feature selection methods, labeled as $m, n, p,$ and $q$, the normalized importance score array for the SNP $i$ is presented as $Z_i = [S_i^m, S_i^n, S_i^p, S_i^q]$. Then, the ground truth score $S_i^{m,n,p,q}$ is generated by taking the harmonic mean of the score array elements as: $S_i^{m,n,p,q} = \frac{4}{(\frac{1}{S_i^m} + \frac{1}{S_i^n} + \frac{1}{S_i^p} + \frac{1}{S_i^q})}$. The harmonic mean balances the contributions of each feature selection method by giving more weights to smaller scores. This balance is pivotal in scenarios where one method produces significantly larger importance scores than the others, preventing any method from overshadowing the overall aggregated score. Additionally, the harmonic mean is robust against outliers Xu (2009).

Using Optuna Akiba et al. (2019), we fine-tuned a three layer neural network for a regression task with the ground truth scores to find the optimal hyper-parameters, including optimizer, learning rate, and dropout. We optimized the network using the mean square error loss function. Then, from the model with the lowest validation loss, the aggregated SNP scores in the output were sorted and the top $K$ SNPs, with $K = (100, 200, 500, 1000)$, were selected. The list of hyper-parameters tuned for the feature selection task are listed in appendix A.1.

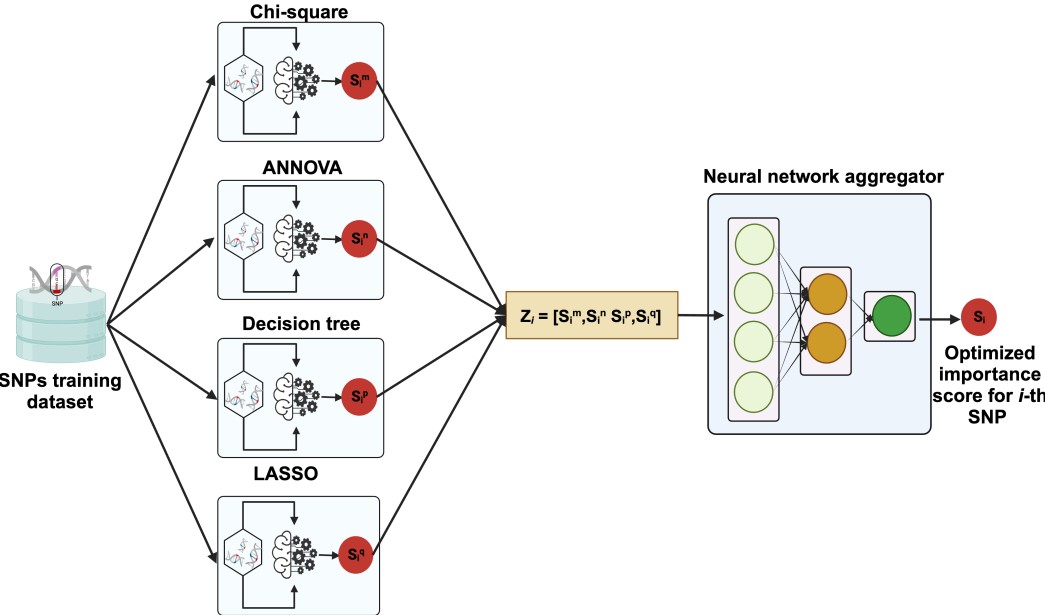

Figure 1: Ensemble-based neural network architecture for combining the output of multiple feature selection methods. In this study, we used Chi-square, ANNOVA, decision tree, and Lasso regression as the feature selection methods.

## 3.2 GRAPH-BASED GENOTYPE DATA REPRESENTATION

Representation learning seeks to transform the initial high-dimensional features into a different feature space, typically composed of non-linear combinations of the original features. To investigate intricate relationships between SNPs, we employ a graph-based representation approach. Graphs offer a valuable advantage by encoding relationships between variables.

Construction of a graph forms the basis of our approach to predict the breast cancer risk. At its core, we construct a graph denoted as $\mathcal{G}$, with nodes representing the individuals as cases or controls, while the filtered $K$-top SNPs, derived from the ENN approach, as the node features. To quantify the similarity between two individuals (nodes), we compute the hamming distance between the two node features. Specifically, the hamming distance $D_{i,j}$ between the two nodes $i$ and $j$ is computed as follows:

$$D_{i,j} = \sum_{k=1}^{K} (x_{i,k} = x_{j,k}).$$

(1)

Here, $x_{i,k}$ and $x_{j,k}$ represent the actual values of the $k-$th SNP for nodes $i$ and $j$, respectively.

The Hamming distance $D_{i,j}$ serves as a metric to quantify the difference between the genotypes of two nodes (individuals). To transform $D_{i,j}$ into a measure of similarity rather than difference, we used the inverse of $D_{i,j}$ as the edge weight in our graph. Specifically, an edge is established between nodes $i$ and $j$ if $D_{i,j} = 0.5$ (See appendix A.2). The weighted graph representation effectively captures the complex interplay of genetic variants, enhancing the ability to predict individual genetic risk for breast cancer with improved accuracy and biological insight.

In this study, we considered three deep learning-based graph architectures, namely, GCN, GAT, and GraphSAGE for a node classification task.

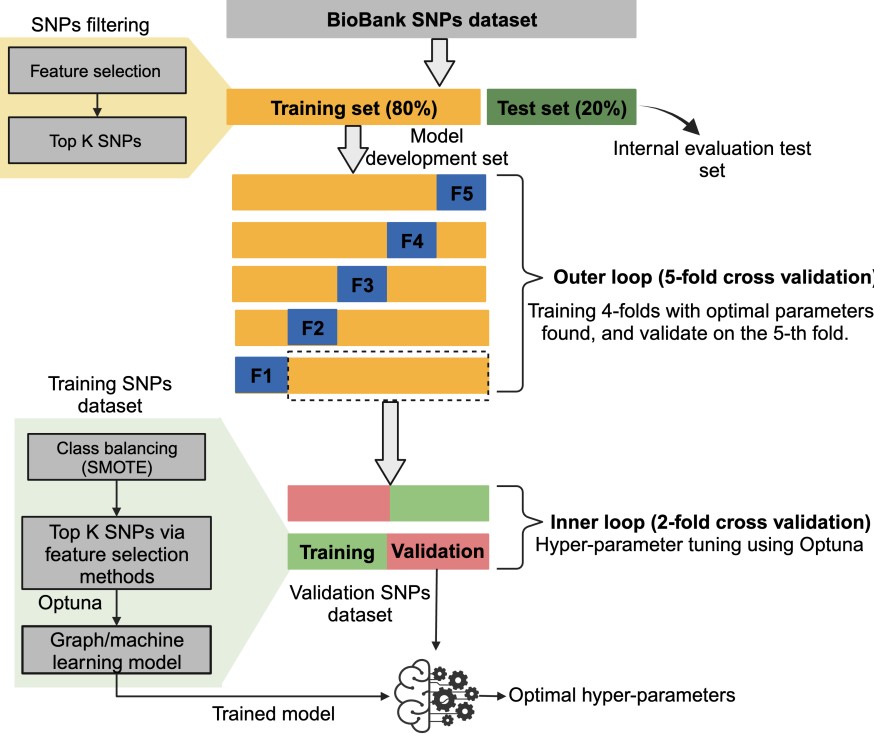

Figure 2: Training and evaluation protocol.

## 4 IMPLEMENTATION DETAILS

Figure 2 illustrates the data splitting protocol employed in this study to train and evaluate the models. The SNPs dataset consists of 1726 individuals (cases: 168, controls:1558), which we divided it into a model development set (80%) (cases:134, controls:1246) and an internal evaluation test set (20%) (cases:34, controls:312) using a stratified splitting protocol. For hyper-parameter optimization and model training, we used nested cross-validation on the model development set. This involved an outer loop (5-fold cross-validation) and inner loop (2-fold cross-validation). To address the class-imbalance issue, we employed the synthetic minority oversampling technique (SMOTE) Chawla et al. (2002) within each training fold, then filtered the SNPs using the ENN model. Within each outer loop training fold, a 2-fold cross-validation was performed for hyper-parameter tuning with Optuna Akiba et al. (2019) (See appendix A.3 and A.4 for list of hyperparameters used for graph models and machine learning classifiers, respectively.). The optimal hyper-parameters found in the inner-loop are used to train the outer loop. Finally, the graph and machine learning model performance was assessed on the internal evaluation test set and metrics reported include precision, recall, accuracy and AUC. We implemented all the graph models using PyTorch geometric (version 2.3.1) Fey & Lenssen (2019) enabled with CUDA (version 11.7), and trained on a NVIDIA Tesla V100 GPU, provided by the CSC-IT Center for Science, Finland.

## 5 RESULTS

### 5.1 SUPERIORITY OF ENSEMBLE-BASED SNP FILTERING OVER ALTERNATIVE FEATURE SELECTION APPROACHES

First, we initiate by evaluating the effectiveness of our newly proposed SNPs selection method, comparing it against well-established statistical techniques such as Chi-squared and ANOVA, as well as machine learning-driven approaches like decision trees and Lasso regression ($L1 = 1$). We used the harmonic mean rank as a baseline for comparison against our proposed ensemble-based

feature fusion method. For each SNP, we computed the ranks from all four methods and calculated the harmonic mean rank, a statistical measure that emphasizes lower ranks, providing a balanced representation of SNP importance across methods.

We chose the initial 100, 200, 500, and 1000 highest-ranked SNPs from each method and employed them to conduct risk prediction on the internal evaluation test set. Increasing the number of SNPs does not guarantee an improvement in prediction accuracy. Figure 3 shows that the optimal outcome is attained using the ENN method, with the top 200 ranked SNPs, yielding an AUC value of 0.986. Among the baseline methods, the Lasso method, employing the 100 top SNPs, achieved the highest prediction accuracy with an AUC value of 0.945.

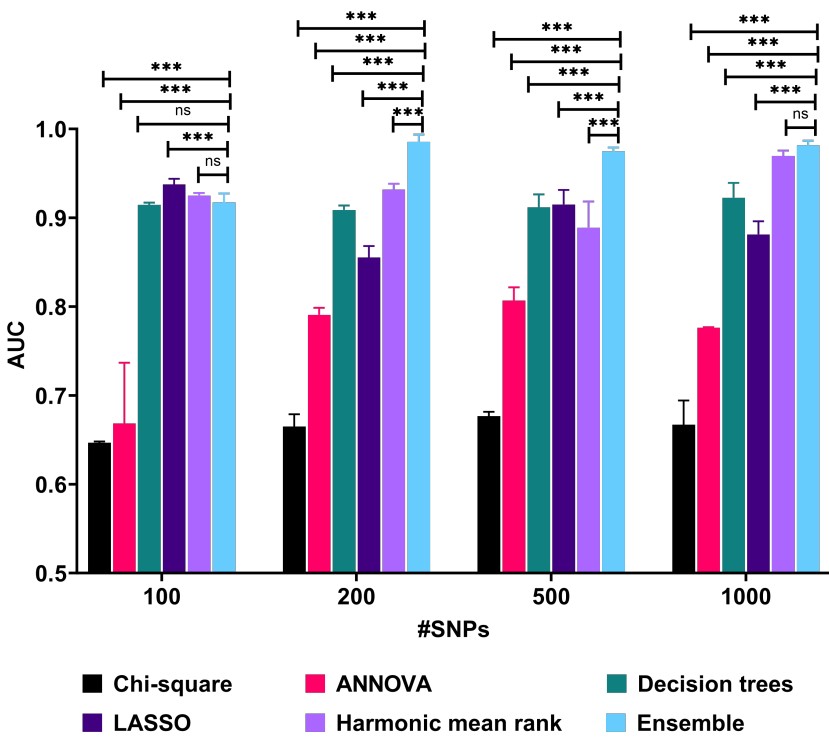

Figure 3: Comparing the performance of multiple feature selection methods across various SNP counts on the internal evaluation test set using the GCN model. An asterisk (***) denotes that the distinction between the ensemble and baseline feature selection methods is statistically significant, as determined by a $t$-test with a $p$-value $\leq 0.001$. ns: not significant.

## 5.2 GCN OUTPERFORMS OTHER GRAPH-BASED METHODS AND CONVENTIONAL MACHINE LEARNING APPROACHES IN PREDICTING THE BREAST CANCER RISK USING THE TOP-RANKED SNPS

Subsequently, employing the top 200 SNPs from the ensemble method, we assess the risk prediction performance of three different graph architectures — namely, GCN, GAT, and GraphSAGE — on the internal evaluation test set in Table 1. GCN demonstrates enhancement, with 1.23% and 2.28% relative increases in AUC compared to GAT and GraphSAGE, respectively. Similarly, the GCN model exhibits substantial improvements in accuracy, with relative increases of 3.79% and 8.24% when compared to the GAT and GraphSAGE models, respectively.

Table 2 illustrates the predictive capabilities of different machine learning techniques, such as XG-Boost, random forest, SVM classifier, logistic regression, and fully connected network (FCN), when applied to the task of breast cancer risk prediction on the test set. These models employ the top-ranked SNPs chosen by the ENN approach. The XGBoost classifier, when applied to the top 500 SNPs, demonstrates superior performance compared to the other machine learning and deep learning classifiers, with an AUC value of 0.955. In contrast to the GCN, machine learning classifiers

Table 1: The performance evaluation of GCN, GAT, and GraphSAGE models for a node-classification task on the internal test dataset. Metrics such as precision, recall, accuracy, and AUC are computed and reported for each model across various SNP counts.

| Model | SNPs count | Precision | Recall | Accuracy | AUC |
|---|---|---|---|---|---|
| GCN | 100 | 0.716 ± 0.006 | 0.835 ± 0.006 | 0.859 ± 0.021 | 0.917 ± 0.010 |
| | 200 | 0.931 ± 0.006 | **0.946 ± 0.006** | **0.958 ± 0.005** | **0.986 ± 0.001** |
| | 500 | 0.875 ± 0.008 | 0.934 ± 0.008 | 0.905 ± 0.002 | 0.971 ± 0.004 |
| | 1000 | 0.891 ± 0.002 | 0.912 ± 0.004 | 0.913 ± 0.012 | 0.975 ± 0.005 |
| GAT | 100 | 0.824 ± 0.009 | 0.939 ± 0.006 | 0.825 ± 0.103 | 0.963 ± 0.002 |
| | 200 | 0.903 ± 0.006 | 0.852 ± 0.006 | 0.923 ± 0.010 | 0.974 ± 0.003 |
| | 500 | 0.912 ± 0.005 | 0.827 ± 0.007 | 0.720 ± 0.214 | 0.954 ± 0.012 |
| | 1000 | 0.816 ± 0.009 | 0.940 ± 0.007 | 0.580 ± 0.123 | 0.966 ± 0.002 |
| GraphSAGE | 100 | 0.775 ± 0.010 | 0.951 ± 0.005 | 0.864 ± 0.004 | 0.931 ± 0.002 |
| | 200 | **0.937 ± 0.006** | 0.897 ± 0.009 | 0.885 ± 0.023 | 0.964 ± 0.002 |
| | 500 | 0.927 ± 0.006 | 0.927 ± 0.005 | 0.928 ± 0.002 | 0.971 ± 0.003 |
| | 1000 | 0.918 ± 0.005 | 0.887 ± 0.003 | 0.914 ± 0.027 | 0.966 ± 0.005 |

exhibited reduced predictive accuracy. The GCN, trained on 200 top-ranked SNPs, outperformed the best XGBoost model with a 3.14% relative AUC improvement.

Table 2: The predictive performance of multiple machine learning classifiers on the test set across various sets of top-ranked SNPs selected by the ENN approach.

| Model | SNPs count | Precision | Recall | Accuracy | AUC |
|---|---|---|---|---|---|
| XGBoost | 100 | 0.906 ± 0.008 | 0.905 ± 0.007 | 0.906 ± 0.008 | 0.948 ± 0.003 |
| | 200 | 0.899 ± 0.007 | 0.915 ± 0.007 | 0.921 ± 0.010 | 0.953 ± 0.001 |
| | 500 | 0.907 ± 0.008 | **0.939 ± 0.008** | **0.929 ± 0.004** | **0.955 ± 0.003** |
| | 1000 | 0.922 ± 0.017 | 0.758 ± 0.012 | 0.914 ± 0.006 | 0.956 ± 0.002 |
| Random forest | 100 | 0.877 ± 0.008 | 0.879 ± 0.009 | 0.797 ± 0.002 | 0.938 ± 0.003 |
| | 200 | 0.881 ± 0.007 | 0.911 ± 0.007 | 0.769 ± 0.015 | 0.944 ± 0.005 |
| | 500 | 0.838 ± 0.005 | 0.916 ± 0.005 | 0.750 ± 0.015 | 0.942 ± 0.005 |
| | 1000 | 0.821 ± 0.008 | 0.869 ± 0.006 | 0.742 ± 0.020 | 0.940 ± 0.003 |
| SVM classifier | 100 | 0.863 ± 0,010 | 0.622 ± 0.010 | 0.655 ± 0.009 | 0.720 ± 0.008 |
| | 200 | 0.898 ± 0.013 | 0.310 ± 0.013 | 0.622 ± 0.005 | 0.768 ± 0.011 |
| | 500 | 0.824 ± 0.010 | 0.912 ± 0.010 | 0.604 ± 0.001 | 0.740 ± 0.005 |
| | 1000 | 0.866 ± 0008 | 0.917 ± 0.005 | 0.591 ± 0.006 | 0.772 ± 0.013 |
| Logistic regression | 100 | 0.828 ± 0.009 | 0.939 ± 0.007 | 0.543 ± 0.011 | 0.587 ± 0.009 |
| | 200 | 0.824 ± 0.009 | 0.937 ± 0.006 | 0.569 ± 0.005 | 0.672 ± 0.012 |
| | 500 | 0.817 ± 0.006 | 0.938 ± 0.006 | 0.566 ± 0.004 | 0.617 ± 0.004 |
| | 1000 | 0.693 ± 0.012 | 0.631 ± 0.011 | 0.512 ± 0.004 | 0.626 ± 0.006 |
| FCN | 100 | 0.714 ± 0.001 | 0.918 ± 0.006 | 0.784 ± 0.008 | 0.908 ± 0.006 |
| | 200 | 0.775 ± 0.010 | 0.931 ± 0.005 | 0.838 ± 0.006 | 0.844 ± 0.004 |
| | 500 | **0.913 ± 0.007** | 0.743 ± 0.011 | 0.836 ± 0.006 | 0.942 ± 0.004 |
| | 1000 | 0.822 ± 0009 | 0.938 ± 0.006 | 0.867 ± 0.006 | 0.941 ± 0.005 |
| GCN | 100 | 0.716 ± 0.006 | 0.835 ± 0.006 | 0.859 ± 0.021 | 0.917 ± 0.010 |
| | 200 | **0.931 ± 0.006** | **0.946 ± 0.006** | **0.958 ± 0.005** | **0.986 ± 0.001** |
| | 500 | 0.875 ± 0.008 | 0.934 ± 0.008 | 0.905 ± 0.002 | 0.971 ± 0.004 |
| | 1000 | 0.891 ± 0.002 | 0.912 ± 0.004 | 0.913 ± 0.012 | 0.975 ± 0.005 |

## 6 DISCUSSION

In this study, we have pioneered the application of graph-based learning techniques to the breast cancer risk prediction task, introducing a pipeline for the representation of SNPs data. Our proposed pipeline encompassed an ensemble-based feature selection approach for SNP filtering and the subsequent creation of a graph-based representation. The potential application of this study is to provide

relevant knowledge of genetic data representation, which can enhance state-of-the-art methods or as an alternative to predict individual's breast cancer risk, conduct risk score analysis, and identify contributing genetic variants.

One of the important points of this study is that we used genetic data from the Biobank of Eastern Finland. This dataset helped us make predictions about breast cancer risk specifically for the Finnish population. However, to make predictions that work well for a broader range of populations, we typically need a lot more genetic data from different groups. In future research, we will test how well our approach performs with other populations.

Regarding finding the optimal number of SNPs, we observed that the prediction accuracy did not improve considerably when the number of SNPs increased or decreased using the graph models. This observation is in line with the observation of the previous machine learning models, where prediction accuracy was not affected by the number of SNPs used as features Behravan et al. (2018); Ho et al. (2020). The best prediction model was obtained using the GCN with 200 top-ranked SNPs selected by our proposed ensemble-based feature selection approach.

Another noticeable observation is that there is a considerable improvement in breast cancer risk prediction accuracy obtained by the GCN compared to the most classical machine learning models. Thus, we can conclude that the graph-based learning can serve as a state-of-the-art approach for predicting individual's breast cancer risk based on genetic variation data.

## 7 CONCLUSION

We advanced the field of breast cancer case-control prediction through the application of graph-based deep learning architectures. The approach we suggest in this study did not rely on picking out specific cancer-related SNPs in advance. Instead, our approach addressed the challenge of SNP filtering by employing an ensemble-based feature selection method, effectively capturing non-linear and high-dimensional SNP-SNP interactions. We compared the power of three distinct graph neural networks, namely GCN, GAT, and GraphSAGE, for breast cancer case-control classification using the top SNPs selected through our proposed feature selection approach. In comparison, we conducted comprehensive experiments with well-established machine learning classifiers like XG-Boost, random forest, SVM classifier, and logistic regression. The GCN, trained on 200 top-ranked SNPs, outperformed the best XGBoost model with a 3.14% relative AUC improvement in predicting the breast cancer risk.

In conclusion, our study introduced a new approach for genotype data representation that leverages graph-based deep learning to enhance breast cancer risk prediction. Further refinements and applications of our approach hold promise for improved breast cancer risk assessment and personalized healthcare.

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

# A  APPENDIX

## A.1  HYPER-PARAMETERS FOR FEATURE SELECTION METHODS

| Task | Algorithm | Paramter | Values [start, end, step] or options |
|------|-----------|----------|--------------------------------------|
| **Feature selection** | Decision tree | maximum depth | [2,32, 2] |
| | | minimum samples split | [0.1, 1.0, 0.2] |
| | | minimum samples leaf | [0.1, 0.5, 0.1] |
| | | criterion | [gini, entropy] |
| | LASSO | alpha | [le-5, 10.0] |
| | Ensemble | learning rate | 1e-5, 1e-1] |
| | | weight decay | [1e-5, 1e-1] |
| | | Optimizer | [Adam, SGD, AdamW] |
| | | number of epochs | [10,1000,10] |
| | | dropout | [0.0, 0.8, 0.2] |
| | | activation | [relu, leaky relu, swish] |

## A.2  OPTIMIZING HAMMING DISTANCE FOR GRAPH NODE CONNECTIVITY

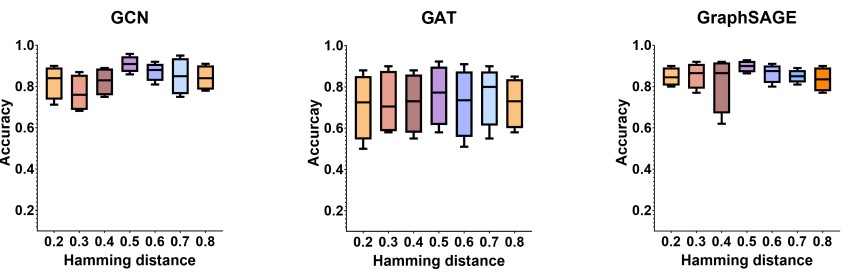

Figure S1: Assessing the performance of GCN, GAT, and GraphSAGE with weighted edge hamming distance across different thresholds. Notably, a $D_{i,j}$ value of 0.5 consistently delivers high accuracy across all graph models.

Figure S1 illustrates the comparative performance of GCN, GAT, and GraphSAGE, across various $D_{i,j}$ thresholds used as a hyper-parameter during model tuning on the development set. The hamming distance threshold serves as a criterion to construct an edge connecting two individuals with the top filtered SNPs as node features. At a hamming distance threshold of 0.5, all the three graph models achieve an optimal accuracy. In specific, at hamming distance of 0.5, the GCN model achieves an accuracy of 0.95, while the GAT and GraphSAGE models reach slightly lower accuracy's of 0.92 and 0.88, respectively. This suggests that the threshold of 0.5 for hamming distance is most effective for the GCN model.

## A.3 Hyper-parameters for graph models

| Algorithm | Hyper parameter | Values [start, end, step] or options |
|---|---|---|
| **GCN** | learning rate | [1e-5, 1e-1] |
| | weight decay | [1e-5, 1e-1] |
| | optimizer | [Adam, RMSProp, SGD, Adagrad, Adadelta, AdamW] |
| | number of epochs | [10,200, 10] |
| | dropout | [0.1, 0.6] |
| | number of layers | [2,4,8] |
| | fully connected hidden dimension | [8,16,32,64] |
| | activation fucntions | [relu, leaky relu, prelu, swish, softplus, sigmoid] |
| | loss fcuntions | [nll loss, cross entropy] |
| **GAT** | learning rate | [1e-5, 1e-1] |
| | weight decay | [1e-5, 1e-1] |
| | optimizer | [Adam, RMSProp, SGD, Adagrad, Adadelta, AdamW] |
| | number of epochs | [10,200, 10] |
| | dropout | [0.1, 0.6] |
| | number of layers | [2,4,8] |
| | fully connected hidden dimension | [8,16,32,64] |
| | activation fucntions | [relu, leaky relu, prelu, swish, softplus, sigmoid] |
| | loss fcuntions | [nll loss, cross entropy] |
| | number of heads | [1,12, 2] |
| **GraphSAGE** | learning rate | [1e-5, 1e-1] |
| | weight decay | [1e-5, 1e-1] |
| | optimizer | [Adam, RMSProp, SGD, Adagrad, Adadelta, AdamW] |
| | number of epochs | [10,200, 10] |
| | dropout | [0.1, 0.6] |
| | number of layers | [2,4,8] |
| | fully connected hidden dimension | [8,16,32,64] |
| | activation fucntions | [relu, leaky relu, prelu, swish, softplus, sigmoid] |
| | loss fcuntions | [nll loss, cross entropy] |
| | aggregator | [mean, sum, max] |

### A.4 HYPER-PARAMETERS FOR MACHINE LEARNING CLASSIFIERS

| Algorithm | Hyper parameter | Values [start, end, step] or options |
|---|---|---|
| **XGBoost** | number of estimators | [50,1000] |
| | maximum depth | [3,10] |
| | learning rate | [0.01, 0.3] |
| | lambda | [1e-5, 10] |
| | alpha | [1e-5, 10] |
| **Random forest** | number of estimators | [50, 1000] |
| | maximum depth | [3, 20] |
| | minimum samples split | [0.1, 1.0] |
| | minimum samples lead | [0.1, 0.5] |
| | bootstrap | [True, False] |
| **Support vector classifier** | kernel | [linear, poly, rbf, sigmoid] |
| | gamma | [scale, auto] |
| | gamma value | [1e-6, 1e-1] |
| **Logistic regression** | penality | [L1, L2] |
| | solver | [liblinear, lbfgs, sage] |
| **FCN** | number of layers | [1,5] |
| | activation fucntions | [relu, leaky relu, prelu, swish, softplus, sigmoid] |
| | loss fcuntions | [nll loss, cross entropy] |
| | learning rate | [1e-5, 1e-1] |
| | weight decay | [1e-5, 1e-1] |
| | optimizer | [Adam, RMSProp, SGD, Adagrad, Adadelta, AdamW] |
| | number of epochs | [10,200, 10] |
| | dropout | [0.1, 0.6] |
| | fully connected hidden dimension | [128, 64, 32, 16, 8, 4] |

