# OpenReview forum: "A GRAPH-BASED REPRESENTATION LEARNING APPROACH FOR BREAST CANCER RISK PREDICTION USING GENOTYPE DATA"
_ICLR.cc/2024/Conference — Submitted to ICLR 2024_

### Official Review · Reviewer_fYgW · 2023-11-01

**Soundness:** 1 poor
**Presentation:** 2 fair
**Contribution:** 2 fair
**Rating:** 3
**Confidence:** 4

**Summary:**

The paper introduces a graph-based pipeline for breast cancer risk stratification, addressing the challenge of high dimensionality and feature interaction in genetic features through an ensemble-based feature selection approach. Based on testing on a dataset from the Biobank of Eastern Finland, it finds that a graph convolutional network (GCN) outperformed classical machine learning models.

**Strengths:**

The paper emphasizes an important problem in dealing with genomic data, namely that of handling SNP interactions and harnessing those to capture disease risk. It employs a GCN-based approach that is, at least in theory, capable of accommodating these constraints via the node features.

**Weaknesses:**

1. *Unclear evaluation metrics*: Could you please clarify exactly how the error bars are computed (say, in Table 2)?
  - For example, in Table 2, a rough calculation for the accuracy error bars for GCN (with 346 test cases) indicates it should be 0.95±0.050, as opposed to the reported 0.95±0.005.
- A slightly different evaluation-related question: it is not clear from the Table 2 results if employing simple FCNs would already provide a good baseline.
2. *Comparisons with extant literature are inadequate; potential lack of novelty*: Please make a clear comparison with literature employing graph-based NN approaches for cancer risk stratification using genomic data. From a methods perspective, it is hard to see the novelty of this work, so my hope was to see a more rigorous comparison with other works in this application area.
- A quick web search yields many potentially relevant papers and it is hard to tell what the differences are.
- Particular for SNP interactions, please consider comparing with [Machine learning identifies interacting genetic variants contributing to breast cancer risk](https://www.nature.com/articles/s41598-018-31573-5)
3. *The graph-based NN approach needs to be explained more clearly*:
- Specifically, how are the node/patient embeddings learned and how does the (Hamming) distance metric come into play here?
- Please clarify whether genomic features are the only features you are using. Are you using any clinical information?
4. *SNP interactions captured?*: How exactly is your approach capturing SNP-SNP and higher-order interactions?
- You say that (paraphrasing) other approaches prune the set of features by considering SNPs one at a time, effectively not consider SNP interactions. However, isn’t that what you are also effectively doing by computing the score S_i for each SNP i?

**Questions:**

Please address the questions/concerns raised in "Weaknesses".

---

> ### Author Response · Authors · 2023-11-19
> **Comparison with other works in this application area.**
>
> We thank the reviewer for this comment. Breast cancer is a multifactorial disease influenced by both genetic and non-genetic risk factors. Risk prediction models utilize these factors to estimate an individual woman's risk of developing breast cancer. One of the challenges in this field is dealing with a large number of Single Nucleotide Polymorphisms (SNPs) (ranging from 10^5 to 10^7) and relatively small sample sizes.
>
> Traditional methods involve standard hypothesis testing where each SNP is individually assessed for its association with the disease based on allele/genotype frequencies. This approach generates p-values that are then adjusted for multiple testing. However, it has limitations as individual SNPs often have a small effect on observed phenotypes, and it overlooks potential correlations and interactions among SNPs.
>
> Regression models are commonly used for cancer risk prediction using a limited set of SNPs, often selected from Genome-Wide Association Studies (GWAS). For instance, some studies have used logistic regression to measure the significance of SNPs, incorporating confounding variables such as sex, body mass index, smoking status, and race [1]. Other approaches include using a gradient tree boosting method to identify groups of SNPs associated with breast cancer risk [2].
>
> Despite the common practice of using logistic regression for risk prediction, there is limited literature on the use of graph-based neural networks (GNNs) for representing genotyped data in cancer risk prediction. However, GNNs have been used in other cancer-related data and tasks. For example, Graph convolutional neural networks have been used in metastasis breast cancer risk prediction, providing patient-specific molecular subnetworks that explain the differential clinical outcome and identify therapeutic vulnerabilities [3]. In another study, a novel approach was presented to predict cancer driver genes using a semi-supervised deep graph learning framework called GGraphSAGE. This method integrates multiomics data, including genomic, epigenomic, and transcriptomic levels, along with protein–protein interaction network-derived attributes. The method was tested on eight tumor types, and the results suggest that GGraphSAGE outperforms several state-of-the-art computational methods for identifying driver genes [4].
>
> To highlight our study contribution, our method introduces an innovative approach to breast cancer risk prediction, diverging from the common practice of using logistic regression and the use of limited number of genetic variants typically employed in such analyses. The novelty of our method lies in the use of ensemble learning techniques and GNNs for interpreting genetic variants and performing classification. We employed ensemble learning that integrates multiple feature selection methods, including both statistical and machine learning-based models. Furthermore, we incorporated a deep neural network that learns to combine weighted feature selections. This interaction enabled us to identify the most predictive SNPs for the task of breast cancer risk prediction. We are confident that our study stands as one of the pioneering efforts in leveraging deep learning methodologies for cancer risk prediction using genetic data.
>
> We will revise the introduction section of the original manuscript to include a review of the work mentioned above, which utilizes GNNs in assessing the cancer risk.
>
>
> [1] Mahesworo B, Budiarto A, Hidayat AA, Pardamean B. Cancer Risk Score Prediction Based on a Single-Nucleotide Polymorphism Network. Healthcare Informatics Research. 2022;28(3):247-255.
>
> [2] Behravan H, Hartikainen JM, Tengström M, et al. Machine Learning Identifies Interacting Genetic Variants Contributing to Breast Cancer Risk: A Case Study in Finnish Cases and Controls. Scientific Reports. 2018;8:13149.
>
> [3] Chereda H, Bleckmann A, Menck K, et al. Explaining Decisions of Graph Convolutional Neural Networks: Patient-Specific Molecular Subnetworks Responsible for Metastasis Prediction in Breast Cancer. Genome Medicine. 2021;13:42.
>
> [4] Song H, Yin C, Li Z, Feng K, Cao Y, Gu Y, Sun H. Identification of Cancer Driver Genes by Integrating Multiomics Data with Graph Neural Networks. Metabolites. 2023;13(3):339.

---

> ### Author Response · Authors · 2023-11-21
> **Please consider comparing with Machine learning identifies interacting genetic variants contributing to breast cancer risk**
>
> Thank you for your suggestion. We have indeed considered the mentioned study. The study proposed an effective machine learning-based approach to identify groups of interacting SNPs that contribute most to the risk of breast cancer. In the study, the researchers adopted a gradient tree boosting method followed by an adaptive iterative SNP search to capture complex non-linear SNP-SNP interactions and consequently, obtained a group of interacting SNPs with high breast cancer risk-predictive potential. They also proposed a support vector machine formed by the identified SNPs to classify breast cancer cases and controls. In our method, we ranked the SNPs using four distinct feature selection methods and addressed the issue of feature redundancy by developing an ensemble-based feature fusion approach. We used a GCN model to predict the risk based on the identified SNPs.
>
> In our study, we used a total of 644,585 SNPs, compared to the 125,041 used in the their study. Both studies employed an additive scheme, representing each SNP based on the count of minor alleles, where homozygous major, heterozygous, and homozygous minor genotypes are encoded as 0, 1, and 2, respectively. Moreover, the dataset used in their study is partly balanced (445 cases, 251 controls), in contrast to our study, which has a significantly imbalanced dataset (168 cases, 1558 controls).
>
> We now replicated the adaptive iterative search, which is the second module of their study, using the top 1000 SNPs in our dataset. The AUC that we obtained was 0.62, a value considerably lower than the 0.986 achieved with our proposed method. This indicates that our approach could potentially offer more precise predictions of breast cancer risk based on SNP interactions. Nevertheless, as the authors of the study pointed out, comprehensive hyper-parameter tuning would be necessary to attain optimal prediction accuracy using their proposed method. We value your suggestion and will continue in investigating and comparing various methodologies in our ongoing work.

---

> ### Author Response · Authors · 2023-11-21
> **How exactly is your approach capturing SNP-SNP and higher-order interactions?**
>
> Thanks for raising this question. Our approach captures SNP-SNP and higher-order interactions through a combination of ensemble-based neural network architecture and graph neural networks (GNNs).
>
> In the first stage, we use Chi-square, ANNOVA, decision tree, and Lasso regression as feature selection methods. Each of the ML methods, unlike the statistical methods, can capture certain types of interactions between SNPs. For example, decision trees naturally model interactions between features through its hierarchical structure. Lasso regression, in other hand, performs a weighted linear combination of the SNPs followed by regularization. An ensemble-based neural network architecture then combines the outputs (importance scores) of all feature selection methods. Each neuron in a layer of the network takes a weighted sum of all its inputs,  allowing it to consider combinations of features. The non-linear activation functions further enable the modeling of complex interactions.
>
> In the second stage, we use a GNN to model the interactions between SNPs. The nodes in the graph represent SNPs, and the edges represent relationships between them. The edge weights are determined by the inverse of the Hamming distance between the genotypes of two individuals, which serves as a measure of similarity. Therefore, a lower Hamming distance (indicating fewer differences in SNPs) results in a higher edge weight, implying higher similarity. Conversely, a higher Hamming distance indicates less similarity and hence a lower edge weight.
>
> During the GNN training, the model learns from both the node features (SNPs) and the connectivity of nodes (interactions between SNPs). This allows our method to capture the interactions among multiple SNPs that affect the disease risk, unlike the traditional GWAS approach that only considers one SNP at a time.

---

> ### Author Response · Authors · 2023-11-21
> **Unclear evaluation metrics: Could you please clarify exactly how the error bars are computed (say, in Table 2)?**
>
> Thanks for the comment. To address the concern regarding the computation of error bars, we employed bootstrap techniques for our evaluation. Specifically, we performed the evaluation on the test set for 100 repetitions with random shuffling. The reported error bars represent the standard deviation of the evaluation metrics across these 100 iterations. For instance, the accuracy of GCN model using top 200 SNPs yielded 0.958 ± 0.005. Here 0.958 is the mean accuracy of the 100 iterations and 0.005 is the standard deviation.
>
> Furthermore, we appreciate the reviewer's interest in the baseline performance of simple Fully Connected Networks (FCNs). In the revised manuscript, we have now included the FCN as one of the baselines. Similar to other baseline models, we employed Optuna framework to find the optimal parameters for the FCN. The FCN model trained with the top 500 SNPs has shown superior performance of 0.913 ± 0.007 in terms of precision when compared to the other baselines, namely XGBoost, random forest, SVM classifier and logistic regression. However, GCN model trained with the top 200 SNPs filtered via the ENN approach has outperformed all the baselines with an average relative improvement of 19.96%, in terms of AUC.

---

> ### Author Response · Authors · 2023-11-21
> **The graph-based NN approach needs to be explained more clearly.**
>
> Thanks for the comment. In our study, we constructed an undirected graph with each individual representing a node with the top 100, 200, 500, and 1000 SNPs filtered by the ENN approach as their node features. We employed genetic similarity metric, quantified by hamming distance to determine the edge connection between the nodes. The hamming distance measures the number of positions at which the corresponding SNP values differ between two individuals. For instance, if two nodes have SNP values that differ in 10 out of 100 positions, their Hamming distance would be 10. To transform this into a measure of similarity rather than difference, we used the inverse of the Hamming distance as the edge weight in our graph. Therefore, a lower Hamming distance (indicating fewer differences in SNPs) results in a higher edge weight, implying higher similarity. Conversely, a higher Hamming distance indicates less similarity and hence a lower edge weight.
>
> The core mechanism of all the graph models, including GCN, GAT, and GraphSAGE is the iterative message passing and feature update process. While training, the node features are updated by aggregating the neighboring nodes considering the edge connections. This update process captures both inter-and intra SNPs interactions and finally updates the node features based on the individual category (case or control). Finally, the classification layer utilizes the updated node embeddings (64 dimensional) to calculate probabilities of nodes belonging to specific classes, typically using softmax. During training, a loss function quantifies the discrepancy between predicted and actual labels for labeled nodes, and optimization techniques, such as Adam, adjust the model's parameters to minimize this loss. This iterative training process enables the graph models to learn meaningful node representations and perform accurate node classification tasks while preserving the underlying graph structure. The optimal hyperparameters used for constructing each graph model is given in Appendix A.2.
>
> Regarding the use of other risk factors than the genetic variants, in our study, we have focused on developing a method that efficiently learns from thousands of genetic variants for the breast cancer risk prediction task. However, we acknowledge the importance of other risk factors and clinical information, and we are exploring its integration in our ongoing study.
>
> In the revised manuscript, we will clarify more clearly how the GNN models are constructed and further elaborate on the use of other clinical factors for the risk prediction in the discussion section.

---

### Official Review · Reviewer_6KR2 · 2023-11-01

**Soundness:** 2 fair
**Presentation:** 3 good
**Contribution:** 2 fair
**Rating:** 3
**Confidence:** 5

**Summary:**

The manuscript "A GRAPH-BASED REPRESENTATION LEARNING APPROACH FOR BREAST CANCER RISK PREDICTION USING GENOTYPE DATA" presents a method to perform feature selection of genetic data to train a graph neural network for breast cancer risk prediction.

In general the study is solid and introduces into state-of the-art from the domain side.

The main novelty of the manuscript is the way the features (SNPs) are selected. In this process the authors apply a three layer neural network and compare it to 4 additional standard methods (Chi-square test, ANOVA, Decision Tree, and LASSO). The ground truth label in training the ensemble neural network is however the harmonic mean of the 4 standard methods.

The selected SNPs are then used with three different graph neural network approaches to predict breast cancer risk.

Overall, the approach is solid, but the novelty is limited, since there have been other graph-based learning approaches for breast cancer risk prediction and the significance of the neural network feature selection is unclear. There are also some problems with the definitions (see below).

**Strengths:**

Very accessible introduction into the genetics field.

**Weaknesses:**

The main novelty of the manuscript is the way the features (SNPs) are selected. In this process the authors apply a three layer neural network and compare it to 4 additional standard methods (Chi-square test, ANOVA, Decision Tree, and LASSO). The ground truth label in training the ensemble neural network is however the harmonic mean of the 4 standard methods. In Fig. 3 the performance of the Ensemble method is compared to the single feature selection tasks. To really judge, whether the ensemble neural network approach is a significant contribution, one would however need a comparison to the SNPs ranked by the harmonic mean described above (since this is the ground truth in all supervised selection tasks). Additionally, there is a problem with the statistical evaluation of significance, since one cannot assume independence of folds over the different cross-validation runs, which would be necessary in order to apply the t-test (it is also not shown that the other assumptions of the test are fulfilled.

The SNPs are then used with three different graph neural network approaches to predict breast cancer risk. Here the performance is not compared to state-of-the-art in breast cancer risk prediction.

Furthermore, the performance metric used does not seem ideal for the task at hand, because there is a high class imbalance and (ROC) AUC can be affected in those scenarios. The authors should at least give additional metrics like area under precision recall curve.

There is also a problem with the definition of the Hamming distance. The way it is written it is a similarity, not a distance. After normalizing (dividing by K), one could turn it into a distance by subtracting it from one, but that is not how it is defined in the manuscript. Furthermore, the D_{i,j}s are used to create the labels for the graph-based approaches and there a threshold of 0.5 is applied, implying that the "normalized" D_{i,j} is used. My recommendation would be to define the distance as described above and then create an edge for the nodes with distance smaller than 0.5. One would just have to define the weights differently.

**Questions:**

My recommendation would be to define the distance as described above and then create an edge for the nodes with distance smaller than 0.5. One would just have to define the weights differently.

Compare performance of ENN to the baseline with the ranking based on the harmonic mean of the four other methods.

Use AUPRC and use appropriate test of significance.

**Details Of Ethics Concerns:**

The researchers are aware that their results are very limited, since they used data from a very homogeneous group of people.

---

> ### Author Response · Authors · 2023-11-15
> **My recommendation would be to define the distance as described above and then create an edge for the nodes with distance smaller than 0.5. One would just have to define the weights differently.**
>
> In response to the reviewer's suggestion, we have now defined the distance between the nodes as described above and created edges for nodes with distances also smaller than 0.5. We have investigated multiple distance thresholds in the range of 0.2 to 0.8. In our experiments, we observed that a threshold of 0.5 consistently yielded optimal performance for all graph-based architectures, including GCN, GAT, and GraphSAGE on the model development set. In specific, at hamming distance of 0.5, the GCN model achieves an accuracy of 0.95, while the GAT and GraphSAGE models reach lower accuracies of 0.92 and 0.88, respectively. This suggests that the threshold of 0.5 for Hamming distance is an optimal choice for the GCN model in this study.
> We will include a new Figure in the revised manuscript which will illustrate the comparative performance of the GCN, GAT, and GraphSAGE models across various Hamming distance thresholds on the development set.

---

> > ### Comment · Reviewer_6KR2 · 2023-11-19
> >
> > If you tested several thresholds in the range of 0.2 to 0.8, was this on additional data (preliminary evaluations)?

---

> > > ### Author Response · Authors · 2023-11-20
> > >
> > > Thanks for the comment. Yes, we have run the experiment on the development set and not on the evaluation set. Please view our updated answer to this comment. We will also include a new Figure on the revised manuscript which show the effect of different Hamming distance thresholds on the accuracy.

---

> ### Author Response · Authors · 2023-11-20
> **Compare performance of ENN to the baseline with the ranking based on the harmonic mean of the four other methods.**
>
> We appreciate the reviewer's suggestion and have conducted an additional experiment accordingly. We used the harmonic mean rank as a baseline for comparison against our proposed ensemble-based feature fusion method.
>
> Initially, we ranked the SNPs using four distinct feature selection methods. Each method assigned ranks to the features based on their importance (as determined by machine learning methods) and relevance scores (as determined by statistical methods). For each SNP, we compiled the ranks assigned by all four methods, resulting in a unique list of ranks for that SNP.
>
> We then calculated the Harmonic Mean Rank (HMR) using these ranks. The HMR is the reciprocal of the average of the reciprocals of the ranks, which assigns more weight to lower ranks, thus providing a balanced representation of the SNP's importance across the methods.
>
> The SNPs were then sorted based on their HMR values in ascending order, and then formed sets of the top 100, 200, 500, and 1000 important SNPs. These SNPs were identified from the development set. A GCN was subsequently trained on these sets, and the prediction accuracy was reported on the evaluation set.
>
> We will update Figure 3 in the main manuscript to include the results of this experiment. The results indicate that the ensemble-based feature fusion significantly outperforms the HMR in all feature sets, except for the set with 100 SNPs. However, the difference in terms of AUC is very small (less than 1%).

---

> ### Author Response · Authors · 2023-11-20
> **Use AUPRC and use appropriate test of significance.**
>
> We appreciate your feedback and have taken it into account. To provide a more thorough evaluation of our model’s performance, in the revised manuscript, we will expand Tables 1 and 2 to include additional metrics such as precision and recall. This is particularly important considering the class imbalance in our dataset.
>
> Our dataset presented a significant class imbalance. To address this, we employed the Synthetic Minority Over-sampling Technique (SMOTE). This method aids in balancing the class distribution, enhancing the model capability to learn from the examples of the minority class. It’s important to note that we only applied SMOTE to the development set.
>
> Furthermore, we have conducted a significance test to compare the performance of various feature selection methods across different SNP counts. We invite you to refer to Figure 3 for this comparison. An asterisk (***) in the Figure indicates a statistically significant difference between the ensemble and baseline feature selection methods. This significance was determined by a t-test with a p-value ≤ 0.001.

---

> ### Author Response · Authors · 2023-11-20
> **The researchers are aware that their results are very limited, since they used data from a very homogeneous group of people.**
>
> We appreciate the reviewer's comment and acknowledge that our study was conducted on a homogeneous population. Not all possible variants are represented in our data. However, it’s worth noting that some variants are more prevalent in this population allowing for a more detailed study of these variants than what might be achievable in other populations. The Finnish population's genetic homogeneity allows for easier identification of specific genetic variants associated with diseases like breast cancer. In a more diverse population, the same disease could be associated with a larger number of different genetic variants, making it more challenging to identify which variants are truly significant.
>
> The Finnish population also exhibits a high degree of linkage disequilibrium, which refers to the non-random association of alleles at different loci. This can be particularly useful in genetic studies, as it can help to identify groups of genes that may collectively influence the risk of developing a disease.
>
> Furthermore, the heterogeneity of breast cancer and its risk factors makes it important to study different populations separately. Some risk factors may be more relevant for some subtypes and populations than others, and the genetic makeup of the Finnish population may reveal unique interactions between these factors.
>
> In this study, our aim was to develop a new method that uses genotyped data and deep learning to find and use interacting SNPs for breast cancer risk prediction. While the prediction accuracy achieved is high, it is specific to this dataset. The model can also be retrained for different datasets and populations to find the best genetic variants for each cohort. We agree with the reviewer that the model needs to be validated in another cohort. To this end, we have applied for a new cohort to validate the model proposed in this study.
>
> We believe that our study, despite its limitations, provides a valuable approach to represent genetic variants for disease risk prediction  and could contribute to the development of risk prediction models in other populations.
>
> We will enhance the discussion section of our manuscript to  incorporate the topic of the population’s genetic homogeneity that was  discussed here.

---

### Official Review · Reviewer_etdw · 2023-11-01

**Soundness:** 1 poor
**Presentation:** 2 fair
**Contribution:** 1 poor
**Rating:** 3
**Confidence:** 5

**Summary:**

This paper performs analyses single neucleotide polymorphism data from a Finnish biobank to predict “breast cancer risk”. Quotations are employed as it’s not clear to this reviewer what the dataset consists of and thus what is actually being predicted (see weaknesses).

The model employs a range of existing methodologies as part of this analysis. This includes an ensemble of feature selection methods, conventional machine learning models and graph convolutional neural networks.

A nested cross validation scheme for hyperparameter selection and model training is employed along with an internal test set for validation.

**Strengths:**

Overall, the application of graph neural networks to this application area is conceptually sensible to model the interactions and correlations between features in SNP data.

Nested cross validation approach makes sense and should yield reliable evaluation results.

**Weaknesses:**

It’s not clear from the manuscript precisely what the samples are – i.e. tissue, blood, etc. – and the nature of the clinical follow-up. The paper states that it is performing “breast cancer risk prediction” however it doesn’t actually describe the dataset itself other than minimal information about case and control numbers.

If indeed the paper is performing risk prediction, I would take this to mean that the genomic samples are all taken from healthy individuals with longitudinal follow-up to establish if they develop breast cancer in the future. There is no details on this given in the manuscript.

The stated results with a ROC AUC of 0.986 is ludicrously high suggesting that something is not correct about the analysis or problem specification.

**Questions:**

End of first paragraph: “Such model such a significant amount of time and have varied limitations”. Please clarify what is meant by significant amount of time and be specific about what the limitations are and which you propose to address.

Please describe the samples which are analysed including the analyte type and the nature of the clinical follow-up. Are there any baseline approaches existing already which can establish risk prediction for this cohort using established risk factors? Examples may include: established risk factors: family history, age, HRT, parity, breast density, BMI, alcohol usage, established genentic risk scores.

---

> ### Author Response · Authors · 2023-11-14
> **Clarify what is meant by a significant amount of time and be specific about what the limitations are and which you propose to address.**
>
> Breast cancer is a complex disease influenced by many factors. Traditional methods for estimating breast cancer risk are slow and need a lot of data and calculations to account for the various factors that influence the disease. Different models use different risk factors, such as age, family history, race/ethnicity, genetic factors, breast density, lifestyle factors, etc. Some models, such as the Gail model [1] and the Tice et al. model [2], use only a few risk factors. The polygenic risk model uses a combination of SNPs but does not account for their interactions. Different models may give different risk estimates for the same woman depending on the selection of risk factors [3]. There is no single best model for predicting breast cancer risk, and different models may suit different purposes and populations.
>
> In our study, our aim was to develop a method to represent the genetic risk factor of the breast cancer. We used an ensemble approach to find the best combinations of SNPs that influence the risk of breast cancer from a large set of 644,585 SNPs. We then used a graph neural network to predict the risk based on the identified SNPs. Our method can capture the interactions among multiple SNPs that affect the disease risk, unlike the traditional GWAS approach that only considers one SNP at a time. Our method is scalable, fast during inference time, and adaptable to other populations.
>
> [1] MH Gail, LA Brinton, DP Byar, DK Corle, SB Green, C Schairer, and JJ Mulvihill. Projecting individualized probabilities of developing breast cancer for white females who are being examined annually. Journal of the National Cancer Institute, 81(24):1879–1886, 1989. ISSN 0027-8874.
>
> [2] Jeffrey A Tice, Steven R Cummings, Rebecca Smith-Bindman, Laura Ichikawa, William E Barlow,
> and Karla Kerlikowske. Using clinical factors and mammographic breast density to estimate
> breast cancer risk: development and validation of a new predictive model. Annals of internal
> medicine, 148(5):337–347, 2008.
>
> [3] Evans D G, Howell A, Breast cancer risk-assessment models. Breast Cancer Research 9, 9(5):213, 2007.

---

> ### Author Response · Authors · 2023-11-14
> **Describe the samples and if there are establish risk prediction for this cohort using risk factors**
>
> Thanks for the comment. Detailed procedures for genotyping, allele calling, and quality control were followed as outlined in Kurki et al. [1]. The data belong to the FinnGen Release 5, which contains data from 224,737 people who had their genotypes checked for quality. Most of them (154,714) used a special Axiom FinnGen1 chip to get their genotypes. The rest (70,023) used other chips that were not made for Finns. A panel of 3,775 Finns was used, who had their whole genomes sequenced with high accuracy (25-30x) to fill in the missing genotypes for the others. The panel had 16,962,023 SNPs and INDELS (small changes in the DNA) that were not too rare (at least 3 copies).  Note that from this genotyped dataset, in this study, we only had access to 644,585 SNPs (after the pre-processing, which is mentioned in our paper) from 168 breast cancer cases and 1558 controls. We selected the controls for this study from people who did not have cancer and who were the same age range as the cases.
>
> We have previously used some established demographic risk factors of the breast cancer, such as factors related to estrogen metabolism and familial history, to combine them with genetic variants using machine learning approaches (the study cannot be cited because of the anonymity policy). We found that the combination of data gave the best accuracy, and genetic factors alone were not enough. Breast cancer is a heterogeneous disease, and so are the risk factors. Some risk factors may be more relevant for some subtypes than others. In this study, we developed a new method that uses genotyped data and deep learning to find and use interacting SNPs for risk prediction. The prediction accuracy we achieved is high, but it is specific to this dataset.  Please note that the Finnish population's genetic homogeneity allows for easier identification of specific genetic variants associated with diseases like breast cancer. However, we need to validate our model in another cohort. The model can also be retrained for different datasets and populations to find the best genetic variants for each cohort. We are currently working on adding other factors, such as breast density, mammographic features, clinical features, and others, to find the best features for risk prediction for each breast cancer subtype. We have also applied for a new cohort to validate the model proposed in this study.
>
> [1] Kurki, M.I., Karjalainen, J., Palta, P. et al. FinnGen provides genetic insights from a well-phenotyped isolated population. Nature 613, 508–518 (2023).

---

### Official Review · Reviewer_VT6g · 2023-11-11

**Soundness:** 2 fair
**Presentation:** 2 fair
**Contribution:** 1 poor
**Rating:** 3
**Confidence:** 4

**Summary:**

The authors presented a graph-based representation learning framework for breast cancer risk prediction using genetic data. They selected informative SNPs by an ensemble approach aiming to capture non-linear and high-dimensional SNP-SNP interactions that aren't possible with linear feature selection approaches, and then used four ML classifiers to evaluate the efficacy of the approach which was compare with graph neural networks. Specifically, the ensemble feature selection combines Chi-square, ANNOVA, decision tree, and Lasso regression.

**Strengths:**

The overall problem was stated clearly, and the introduction in Section 1 was well-written. The approach itself and the figure illustrations made sense and helped readers understand the material.

**Weaknesses:**

I believe additional evaluation is necessary for both feature selection and risk prediction tasks (i.e., assessing performance with external datasets and comparing results with existing methods). The significance of the proposed approach remains unclear without sufficient validation and justification. I am uncertain about the extent of the method's transferability to other datasets.

There is a lack of interpretation of the results, e.g., the selected top SNPs. No evaluation or comparison was made for those predictive SNPs. The selection of the number of top SNPs also appears a bit arbitrary although the authors claim it didn't affect the results much.

It is not clear how the proposed feature selection pipeline captures higher-order SNP relationships.

The novelty of the paper seems to be limited as other reviewers mentioned from a methodological point of view.

Certain sections of the paper are unclear, please see my questions below.

**Questions:**

1. I don't quite understand how the Hamming distance measures similarity here. If the top SNPs are entirely different between two nodes, even if they have similar values (as coded in 0, 1, 2), it doesn't necessarily mean they are similar. Could the authors clarify this measure?

2. Did the authors control for independence between the training and testing sets, given that they were from the same cohort?

3. In Figure 3, I observe 'top 1000' as the top performer for the ensemble method instead of 'top 200'. The bar doesn't appear to represent a value of 0.986. Could the authors clarify?

4. In Section 2, did the authors intend to say 'SNPs with missing variants greater than 5%' and 'kept the SNPs with linkage disequilibrium of r2 < 0.6'? Normally, people perform LD pruning to reduce the number of SNPs in high LD for feature selection, rather than keeping them.

5. In Section 5.1, which method was used for prediction?

---

> ### Author Response · Authors · 2023-11-14
> **Did the authors control for independence between the training and testing sets, given that they were from the same cohort?**
>
> Thanks for the comment. Yes, the training and test sets are independent and do not overlap. To create the independence between the training and testing sets, we used a stratified splitting protocol to divide the SNPs dataset into a model development set and an internal evaluation test set, maintaining the same proportion of cases and controls in each set as in the original dataset. Furthermore, we used nested cross-validation on the model development set, which is a technique that can prevent data leakage and improve the generalization ability of the model.
> In summary, we used the model development set for tuning the hyper-parameters, selecting the features, and training the model, and then evaluated the performance of the graph model on the internal evaluation test set. As a future direction, we plan to test our model on an external genotyped dataset, for which we have applied to access the data.

---

> ### Author Response · Authors · 2023-11-15
> **I don't quite understand how the Hamming distance measures similarity here. If the top SNPs are entirely different between two nodes, even if they have similar values (as coded in 0, 1, 2), it doesn't necessarily mean they are similar. Could the authors clarify this measure?**
>
> Thank you for your comment regarding the application of Hamming distance in our graph network analysis, where each node represents an individual, and node features are the genotype variants encoded as 0, 1, or 2.
>
>
>
> In our methodology, the Hamming distance serves as a metric to quantify the difference between the genotypes of two individuals. Specifically, it measures the number of positions at which the corresponding SNP (Single Nucleotide Polymorphism) values differ between two nodes. For instance, if two nodes have SNP values that differ in 10 out of 100 positions, their Hamming distance would be 10.
>
> To transform this into a measure of similarity rather than difference, we used the inverse of the Hamming distance as the edge weight in our graph. Therefore, a lower Hamming distance (indicating fewer differences in SNPs) results in a higher edge weight, implying higher similarity. Conversely, a higher Hamming distance indicates less similarity and hence a lower edge weight.
>
> Regarding your concern about the top SNPs being entirely different between two nodes: our method is based on the premise that genetic similarity can be inferred from the overall pattern of SNP variants, rather than the presence or absence of specific top SNPs. We hypothesized that even if the top SNPs differ, a high degree of overall genetic similarity (reflected by a low Hamming distance and thus a high edge weight) can still indicate a meaningful biological relationship. Moreover, we have also calculated the individual feature weights (SNP importance) for each node in the internal evaluation test set using the GNNExplainer method. To further explore the biological roles of the top SNPs that were identified as important features by the GNN model, we are conducting a follow-up study that will include SNP-gene interaction analysis and other methods to make clear how these SNPs affect the pathways related to breast cancer and its subtypes.
>
> Furthermore, we set a threshold for edge formation in the network: an edge exists between nodes if the edge weight (inverse Hamming distance) is greater than 0.5. This threshold would make nodes with sufficiently high genetic similarity be connected, which may enhance the biological relevance of the network structure.
>
> We believe this approach, while abstracting from specific SNP identities, captures a broader and more comprehensive picture of genetic similarity. It is particularly useful in studies where the focus is on understanding the overall genetic relationship patterns within a population, rather than the effects of specific SNPs.
>
> We hope this explanation clarifies the use of Hamming distance in our study. We will include a more detailed explanation of this method in our revised manuscript.

---

> ### Author Response · Authors · 2023-11-15
> **In Figure 3, I observe 'top 1000' as the top performer for the ensemble method instead of 'top 200'. The bar doesn't appear to represent a value of 0.986. Could the authors clarify?**
>
> Thank you for your observation regarding the performance values depicted in Figure 3 of our original manuscript. We have identified a typographical error in the representation of the performance values for the ensemble method.
>
> Specifically, the value for the 'top 200' performer was incorrectly typed as 0.968 instead of the correct value of 0.986. This typographical mistake led to the apparent inconsistency you noted in the figure. We appreciate your attention to this.
>
> We assure you that this error does not reflect any inaccuracies in our data or analysis but is purely a mistake in the figure. In the revised version of our manuscript, we will correct this issue in Figure 3 to reflect the performance value of 0.986 for the 'top 200' ensemble method.

---

> ### Author Response · Authors · 2023-11-15
> **In Section 2, did the authors intend to say 'SNPs with missing variants greater than 5%' and 'kept the SNPs with linkage disequilibrium of r2 < 0.6'? Normally, people perform LD pruning to reduce the number of SNPs in high LD for feature selection, rather than keeping them.**
>
> Thanks for your attention. In Section 2 of our manuscript, we apologize for any confusion in our description. The typo regarding the statement 'kept the SNPs with linkage disequilibrium of r 2 ≥ 0.6' was in the manuscript and not in the actual model development. The PLINK software we used for LD pruning indeed ensures the reduction of SNPs in high LD (r 2 <= 0.6), as per common practice. We appreciate the reviewer's attention to detail, and we will revise the manuscript to accurately reflect our methodology and the use of PLINK for LD pruning."
>
> We acknowledge the typo in the manuscript, however, this has not affected the model development, and we reassure the reviewer about the correctness of the methods and the results.

---

> ### Author Response · Authors · 2023-11-15
> **In Section 5.1, which method was used for prediction?**
>
> In Section 5.1, Figure 3, we used the Graph Convolution Network (GCN) for the case-control prediction task. In this graph, each individual (sample) is represented as a node, and the filtered SNPs serve as the node features. To establish connections between nodes, we calculated the Hamming distance between them. An edge was created between two nodes if the Hamming distance exceeded 0.5.
>
> For more detailed information regarding the architecture and hyperparameters used for the graph construction, please refer to Appendix Section A.2 of the manuscript.

---

### Meta-Review · Area_Chair_sYNs · 2023-12-10

**Metareview:**

The paper presents an interesting approach to breast cancer risk prediction using genetic data, specifically focusing on the selection and analysis of single nucleotide polymorphisms (SNPs). The authors propose an ensemble feature selection method combining Chi-square, ANOVA, decision tree, and Lasso regression to capture complex SNP-SNP interactions. This is followed by the application of graph neural networks (GNNs) for risk prediction.

The paper addresses a significant problem in genomics and employs a theoretically sound approach using graph-based representation learning. The introduction and problem formulation are clear and well-articulated.

The paper lacks clarity in several aspects, including the specifics of the dataset used (sample types, clinical follow-up details), which is crucial for understanding the context and applicability of the findings. The evaluation of the proposed method is insufficient. There is a need for external validation, comparison with existing methods, and a more detailed analysis of the selected SNPs. The novelty of the approach is questioned, given the existence of other graph-based learning approaches in this domain. The method's ability to capture higher-order SNP relationships and the rationale behind the selection of top SNPs are not adequately explained. The use of ROC AUC as the sole performance metric is questionable, especially given the class imbalance in the dataset. Additional metrics like the area under the precision-recall curve should be considered. The paper has statistical evaluation issues, particularly regarding the independence of folds in cross-validation and the assumptions of the t-test used. The definition and application of the Hamming distance in the context of the study are not clear and potentially flawed.

While the paper tackles an important problem and proposes an interesting approach, there are significant issues with clarity, evaluation, and novelty that need to be addressed. The lack of detailed dataset description, insufficient validation, and unclear methodological explanations significantly hinder the paper's impact. Therefore, the recommend is a rejection of this submission in its current form. However, with substantial revisions addressing the concerns raised, particularly around dataset specifics, evaluation robustness, and methodological clarity, this work could potentially contribute meaningfully to the field.

**Justification For Why Not Higher Score:**

The paper lacks clarity in several aspects, including the specifics of the dataset used (sample types, clinical follow-up details), which is crucial for understanding the context and applicability of the findings. The evaluation of the proposed method is insufficient. There is a need for external validation, comparison with existing methods, and a more detailed analysis of the selected SNPs. The novelty of the approach is questioned, given the existence of other graph-based learning approaches in this domain. The method's ability to capture higher-order SNP relationships and the rationale behind the selection of top SNPs are not adequately explained. The use of ROC AUC as the sole performance metric is questionable, especially given the class imbalance in the dataset. Additional metrics like the area under the precision-recall curve should be considered. The paper has statistical evaluation issues, particularly regarding the independence of folds in cross-validation and the assumptions of the t-test used. The definition and application of the Hamming distance in the context of the study are not clear and potentially flawed.

**Justification For Why Not Lower Score:**

N/A

---

### Decision · Program_Chairs · 2024-01-16

Reject